# Effects of Multifaceted Determinants on Individual Stress: The Mediating Role of Social Capital

**DOI:** 10.3390/ijerph19095571

**Published:** 2022-05-04

**Authors:** Chia-Yuan Yu, Kenneth Joh, Ayoung Woo

**Affiliations:** 1School of Public Administration, University of Central Florida, Orlando, FL 32801, USA; ychiayuan@gmail.com; 2Department of Transportation Planning, Metropolitan Washington Council of Governments, Washington, DC 20002, USA; kjoh@mwcog.org; 3Graduate School of Urban Studies, Hanyang University, Seoul 04763, Korea

**Keywords:** social capital, mediating effect, stress, physical environment, community revitalization

## Abstract

Stress substantially results in various negative health outcomes. While there is a nexus between social capital and individual stress, previous studies have primarily explored the direct relationship between them. Social capital may potentially have an indirect effect on perceived stress via social networking pathways that provide accessible resources. This study addresses this research gap by exploring the mediating effect of social capital for associations between personal-level features, personal-level behaviors, physical environments, and perceived stress. A household drop-off survey of 600 respondents was collected from two neighborhoods in Korea and analyzed by structural equation models. Results showed that social capital acted as a mediator on perceived stress level. The frequency of community center use had both direct and indirect impacts on stress level through social capital. Those who were satisfied with the cleanliness of the neighborhood had a higher level of social capital and a lower level of stress indirectly through social capital. Households with more children had a lower level of social capital, while persons who had chronic disease and were more extroverted, agreeable, and open to others enjoyed a higher level of social capital. The results provide policy implications on how community revitalization affects social capital and perceived stress.

## 1. Introduction

It is well-known that stress is strongly correlated with a variety of negative health outcomes such as poor sleep quality, weakened immune system, and decreased cardiovascular health [1,2]. Persons with a high level of perceived stress are more likely to have health-risk behaviors including physical inactivity, unhealthy eating habits, and daily smoking [3,4]. Persons with high stress levels are also prone to increased depression [5].

Physical and neighborhood environments have the potential to influence mental health and perceived stress [6]. Environmental features such as green spaces, parks, accessibility to amenities, perceived safety, walkability, aesthetics, and maintenance of community facilities are significant factors [6,7,8,9]. Furthermore, social environments in neighborhoods could also enhance mental health and alleviate depression. Particularly, social capital has been widely defined as empirical measures of social environments that are representative of social networks, participation, and trust among community residents [5,10,11]. For example, social networks and trust identify what residents feel in social environments, while social participation refers to what residents do in social settings. Healthy and sustainable social environments can be fostered if residents are willing to participate in various community activities, thereby building trust and extending networks with their neighbors [12,13,14,15]. Hence, persons living in an environment with a high level of social capital have been associated with positive health outcomes and behaviors [6].

While there is a nexus between social capital and perceived stress, previous studies have primarily explored the direct relationship between them [16,17,18]. However, it is anticipated that social capital acts as a mediator between multiple factors (e.g., socio-demographic characteristics, personal factors, neighborhood environments, etc.) and stress level. Social capital may potentially have an indirect effect on perceived stress via social networking pathways that provide accessible resources. Given the literature supporting this potential indirect mechanism, it is imperative to extend our knowledge to explore the mediating role of social capital.

This study addresses the following research questions: what multifaceted factors influence social capital, and how do these factors in turn affect individual perceived stress? To answer these questions, we focused on two neighborhoods (Samduck and Sansae communities), implemented by the Residential Environment Improvement Project (REIP) in Seoul, Korea. The REIP aims to revitalize communities in terms of improving deteriorated physical environments and enhancing social capital by constructing the community center. After its implementation in 2012, Samduck and Sansae communities have become representative of successful REIP sites. Samduck and Sansae communities are located in the northern and northwestern areas of Seoul, respectively, which consist of older and distressed communities. Based on a survey of 600 respondents in these project sites, we develop a conceptual framework to examine the mediating effect by employing a structural equation model (SEM). Our findings show that multifaceted factors are associated with perceived stress levels through the mediating factor of social capital. The frequency of community center use has both direct and indirect impacts on stress levels through social capital. Respondents who are satisfied with the cleanliness of the neighborhood have a higher level of social capital and a lower level of stress indirectly through social capital. Additionally, respondents who have chronic disease and are more extroverted, agreeable, and open to others enjoy a higher level of social capital, while those with more children have a lower level of social capital. The findings of this study may help planning practitioners and policymakers better understand how various attributes in the revitalized community affect social capital and perceived stress.

## 2. Literature Review

This study builds upon the literature examining the structural associations among individual stress level, social capital, and their multi-explanatory factors. To identify the structural frameworks and conduct empirical analyses, this study comprehensively reviews empirical findings of previous literature; a careful review of the literature shows that multifaceted domains (socio-demographic characteristics, personal health status, residential behavior characteristics, individual personality, and physical environments) may affect individual perceived stress, and social capital may be a mediator between them.

Stress has been recognized as a cognitive process during a circumstance when resources are perceived as insufficient [19]. Perceived stress could be approached from several aspects and mechanisms between these perspectives [1]. The social structure where individuals are located is highly correlated with their exposure to stress, associated stressors, perception of available resources, and health outcomes [20]. Consequently, socio-demographic characteristics may enable or constrain the available resources for individuals to cope with their perceived stress [21]. Females, unmarried persons, persons with a lower education level, and racial/ethnic minorities were found to have poorer access to available resources and higher exposure to stress [1,3]. Therefore, these socio-demographic groups may find different pathways to perceive stress.

Household characteristics also play an essential role in affecting health status. These characteristics represent an individual’s social connections and networks in forming their status and health conditions since most people spend the majority of their time at home [22]. Therefore, persons from the same household may experience similar tendencies of health-related outcomes [23].

Physical environments are critical factors in affecting perceived stress [24]; they are associated with psychological well-being and mental health [25]. The neighborhood where people live represents the availability and accessibility of the infrastructure, community space, health service, and social networks associated with perceived stress and health outcomes [3]. The resources in the community setting (e.g., parks, recreational spaces, community center, etc.) could serve as places where there are opportunities for social interactions and promote neighborhood cohesion and social connections [7]. However, if neighborhoods are rife with disorder and violence, inadequate housing, and lack of amenities, they may serve as stressors [25]. Previous studies have shown that people in deprived communities have a higher level of stress than those in wealthier neighborhoods due to its limited community services, high crime rates, inadequate transportation and infrastructure, and a lack of social support [3,25,26,27]. In contrast, neighborhoods with a high quality of aesthetics, perceived safety, and social cohesion were associated with lower levels of perceived stress [6].

Additionally, social capital plays an essential role in influencing perceived stress [28,29]. Social capital could be defined as a social network relationship within a group of people [30]. Putnam [11] defined social capital in five critical aspects: (1) community and personal networks; (2) civic networks and participation; (3) local networks (i.e., sense of belonging); (4) reciprocity and norms of cooperation; and (5) community trust. Social capital could also be categorized into cognitive and structural dimensions. Cognitive dimension represents community perception such as community and personal network, reciprocity, trust, and sense of community, while structural dimension emphasizes the behaviors of residents such as their participation in community activities [31]. Based on this definition, the sense of community could be one aspect of social capital. Previous studies have demonstrated that social capital could reduce the negative externalities related to stress [32,33] through various pathways such as providing social support, offering access to available resources, and promoting self-esteem [34]. However, low-income communities typically have a lower level of social capital because low-income neighborhoods tend to have unsafe environments, which hinders social interactions with neighbors [6].

It is expected that social capital would act as a mediator between multifaceted factors and stress level. For instance, residents who use the community center more frequently may have a higher level of social capital and community belonging, which in turn affect their individual stress level. Duration of residence in the community may also influence social capital and stress level. Thus, social capital may indirectly affect perceived stress through social networking pathways that deliver feasible resources such as access to local facilities and program participation [33]. However, at least to our knowledge, most previous studies tested the direct relationship between social capital and mental health [16] and few studies have explored the possible mediating effects of social capital on stress level. Hence, it is necessary to explore the complex associations and underlying mechanisms between social capital and perceived stress.

Despite consistent findings on the significant effects of social capital on perceived stress, it is crucial to explore what factors influence social capital. Previous studies have demonstrated that individual and household characteristics are critical determinants of social capital [16,35]. It would be expected that an individual’s level of trust and network with others would be shaped by the existing household networks; the willingness to participate in civil society would also be shaped by other family members [33]. 

In terms of physical environments, amenities such as green spaces or parks have been identified as a crucial factor related to social capital. Green spaces serve as the shared places for community residents to interact with each other and engage in social activities [36]. Having easy access to public parks or green spaces near the place of residence could promote physical activity, foster social support, and reduce mental stress [36,37]. Furthermore, the perception of safety is significantly associated with willingness to participate in community activities and interact with neighbors [7]. The perception of crime safety also affects the desire of residents to walk in the community [38]. Moreover, neighborhoods with a high walkability index could promote social capital by encouraging people to walk in the community and increase opportunities to interact with neighbors and to develop a strong sense of community [7]. Neighborhood maintenance and overall cleanliness were also identified as critical environmental features associated with social capital [39].

Furthermore, most studies were conducted in the context of Western societies. Few studies have explored the association between social capital and perceived stress in the Asian context. A recent study from Korea [16] examined the relationship between social capital and perceived stress and found a negative correlation. Norstrand and Xu [29] investigated the association between social capital and physical and emotional health among older adults in both urban and rural settings in China and found a significant relationship in urban settings. However, these studies only tested the direct relationship between social capital and stress and did not explore the multifaceted dimensions of these associations. Since Asian cities have experienced rapid growth and dramatic change in terms of population and urban development in recent decades [40], it is crucial to understand how urban and community development influence social capital and perceived stress in order to monitor the development process in developing countries.

## 3. Analytic Methods

Based on the literature review, this study developed the following conceptual framework. The framework includes variables from socio-demographic characteristics, personal health status, residential behavior characteristics, individual personality, physical environments, social capital, and health-related outcomes (e.g., individual stress level). This study hypothesizes that (1) socio-demographic characteristics, personal health status, residential behavior characteristics, individual personality, and physical environments have direct associations with social capital and health-related outcomes, and (2) social capital also acts as a mediator between these domains (socio-demographic characteristics, personal health status, residential behavior characteristics, individual personality, and physical environments) and health-related outcomes (Figure 1).

### 3.1. Study Area, Population, and Data

The study area is two neighborhoods (Samduck and Sansae communities) in Seoul, Korea. Seoul Metropolitan Government has implemented a community revitalization program called the Residential Environment Improvement Project (REIP) in several neighborhoods to create a healthy community. After Seoul Metropolitan Government implemented the REIP in several neighborhoods to promote healthy and socially sustainable communities, Samduck and Sansae communities emerged as the shining examples of successful community developments. Because Seoul Metropolitan Government officials consider the community center as a core facility to enhance social capital in the neighborhood, the substantive outcome of REIP sites is to build a community center managed by community residents (Figure 2). Furthermore, during the REIP development process, residents could participate in the planning process by selecting the location of the community center. Despite numerous efforts building community centers for residents, there is limited understanding of whether community centers encourage the formation of social capital and enhance health outcomes.

In order to explore the factors related to social capital and stress level after the implementation of the community center, a survey was conducted to collect information on the respondent’s stress level, social capital, socio-demographic characteristics, personal health status, individual personality, and physical environmental features. The survey for this study involved humans (i.e., residents) living in REIP sites; hence, the survey was approved by the Institutional Review Board of Hanyang University to identify ethical clearance (Ref. No. HYUIRB202203005). A professional survey firm (Hankook Research Corporation, Seoul, Korea) by request of the Seoul Metropolitan Government employed the household drop-off approach to deliver the questionnaire in person and conducted the face-to-face interview during weekdays from October to November 2020. A random sampling method was used to interview adults, aged 19 and older, who have lived in the Samduck and Sansae communities for more than one year. Trained interviewers delivered the questionnaires to respondents and provided guidance in filling out the survey for household participants to increase the survey’s accuracy and response rate. In response to the COVID-19 pandemic, facial masks were provided to all respondents as an incentive and to confirm that respondents were Samduck and Sansae residents. In total, 600 households were recruited and completed the survey; 17 surveys were excluded due to missing information. Therefore, the final sample size is 583 (417 from Samduck and 166 from Sansae) and the response rate is 97.17%. The very high response rate shows the effectiveness of the drop-off approach and the strong interest of residents on the REIP development. From the sample, the average age of respondents is 54.99; 26.76% of them are male, and 53.86% of them owned their residence.

### 3.2. Variables and Measurements

Table 1 shows the domains, variables, measurements, and descriptive statistics. For stress level, this study asked respondents the daily stress level ranging from 1 (no stress) to 4 (very stressful).

Social capital has been widely defined as the resources available to people through social networks [5,10]. Social capital could be considered as social network relationships within a group of people [30]. Social capital has been widely identified in social, planning, and public health research as a conceptual factor to estimate social network relationships, while the measurements of social capital vary across research disciplines [41]. However, this study builds on a body of literature assessing social capital, particularly in terms of network, participation, sense of belonging, reciprocity, and community trust [5,15]. For social capital, this study generated a latent variable by using nine survey items (e.g., have many conversations with neighbors, keep in touch with neighbors, neighbors to ask for advice, neighbors could be counted on to help when in trouble, trust neighbors, think neighbors trust each other, participate in community activities or volunteer services, and have a sense of community in the neighborhood) ranging from 1 (never) to 4 (always). 

Socio-demographic characteristics include age, gender, household monthly income, marital status, homeownership, education level, number of household members, number of children, and region. Personal health status includes chronic disease, regular exercise, enough sleep, regular meal, non-drinking life pattern, and non-smoking life pattern. For residential behavior characteristics, this study considers living duration, walking frequency, and the frequency of community center use. This study generated a latent variable on individual personality by including nine survey items ranging from 1 (strongly disagree) to 5 (strongly agree). Based on the Big Five model of personality, this study identified individual personality in terms of extroversion, openness, and agreeableness [42]. For physical environments, this study measured the respondent’s satisfaction on walkable neighborhood, transit accessibility, amenity, crime safety, and cleanliness by using an ordinal scale from 1 (not satisfied) to 4 (very satisfied).

### 3.3. Data Analysis

Descriptive statistics were used to explore the mean, standard deviation, and distribution values of all variables within the domains of socio-demographic characteristics, personal health status, residential behavior characteristics, individual personality, physical environments, social capital, and stress level (Table 1).

A structural equation model (SEM) was used to explore the hypothesized conceptual framework; this study examined the mediating effect of social capital on the stress level and the identified multifaceted factors related to social capital and stress level. This study first established the measurement model to create the latent variable for social capital and individual personality. The construction of latent factors was based on theoretical survey design and empirical factor loadings from factor analyses. Second, the structure model was generated to explore the hypothesized relationships among all variables. Maximum Likelihood (ML) was employed as the default estimator. Assessment of the model’s fit was based on the Comparative Fit Index (CFI), Tucker–Lewis Index (TLI), and Root Mean Square Error of Approximation (RMSEA). The model was deemed acceptable when the RMSEA was less than 0.05 and CFI and TLI were over 0.90 [43]. The SEM model was developed and reported with the estimated results of standardized coefficients in order to compare the relative influence of each independent variable on outcomes. M-Plus 8.5 was used to conduct the above analyses.

## 4. Results

This study used SEM to test the hypothesized relationships based on the proposed conceptual framework (Figure 3). Figure 3 shows the overall results for examining the nexus between social capital and perceived stress; particularly, social capital acted as a mediator between multiple factors (e.g., socio-demographic characteristics, personal factors, neighborhood environments, etc.) and stress level. The RMSEA for the model was 0.035 (<0.05 indicating a good fit); the CFI was 0.91 (>0.90 indicating a good fit); and the TLI was 0.91 (>0.90 indicating a good fit). The result showed that respondents with a high level of social capital (standardized coefficient = −0.536, *p* < 0.01) had a lower stress level.

Table 2 shows the relationships between socio-demographic characteristics and two outcomes (social capital and stress level). Households with a higher income level (standardized coefficient = −0.206, *p* < 0.01) had a lower level of social capital. Moreover, households with more children (standardized coefficient = −0.135, *p* < 0.05) had a lower level of social capital. No socio-demographic variables were significantly associated with perceived stress level.

Table 3 shows the direct and indirect relationships between personal health status, residential behavior characteristics, social capital, and stress level. For personal health status, those who had chronic disease (standardized coefficient = 0.284, *p* < 0.01) had a higher level of social capital. Respondents who used the community center more frequently had a higher level of social capital (standardized coefficient = 0.277, *p* < 0.01). For the perceived stress level, those who engaged in regular physical exercises (standardized coefficient = −0.106, *p* < 0.05) had a lower stress level. Respondents with longer living duration in the neighborhood (standardized coefficient = −0.229, *p* < 0.01) had a lower stress level. Respondents who used the community center more frequently had a lower stress level (standardized coefficient = −0.500, *p* < 0.01). In terms of the indirect impact on stress level, the use frequency of the community center had an indirect impact (standardized coefficient = −0.148) on stress level through social capital. However, the direct impact was stronger than the indirect influence (−0.500 > −0.148). Those who had chronic disease were associated with a lower stress level through social capital (standardized coefficient = −0.152).

Table 4 shows the direct and indirect relationships between individual personality, physical environments, social capital, and stress level. Those who were more extroverted, agreeable, and open to others (standardized coefficient = 0.090, *p* < 0.01) had a higher level of social capital. Furthermore, those who were satisfied with the cleanliness of the neighborhood (standardized coefficient = 0.143, *p* < 0.01) had a higher level of social capital. No physical environment factors were directly significantly associated with stress level. The satisfaction of cleanliness in the neighborhood was associated with the stress level indirectly through social capital (standardized coefficient = −0.077). Respondents who had extroverted, agreeable, and an open personality also had a lower stress level through social capital (standardized coefficient = −0.048).

## 5. Discussion

This study explored the mediating role of social capital, especially in terms of network, participation, reciprocity, trust, and sense of community, on stress level. This research specified what factors were related to social capital, and how it in turn affected the perceived stress. Based on these findings, among significant factors associated with stress level, social capital was the most substantial factor with the largest standardized coefficient (−0.536). Therefore, examining how to enhance the level of social capital plays an important role in reducing stress level.

Furthermore, this study illustrated that social capital was influenced by several identified factors, a finding that could help develop tailored and specific policy implications to enhance community belonging. We highlight the following three recommendations to promote social capital and reduce stress level.

First, physical environments play a crucial role in enhancing social capital and reducing stress level. Although the cleanliness of the neighborhood did not directly impact stress level, it could indirectly lower the stress level by enhancing the level of social capital. Therefore, community governing board members and residents should work together to encourage neighborhood cleanliness to promote social capital and reduce stress level. Community-based programs to engage in community activities including neighborhood cleanliness and maintenance could strengthen feelings of community belonging and thereby reduce stress level.

Second, the frequency of community center use affected the stress level directly and indirectly through social capital. This illustrates the importance of promoting community center use as a strategy to foster community-level health. Developing events or programs at the community center to increase community center use should be a priority for the community committee. Engaging in group-based activities could build trust, reciprocity, network, and enhance participation, which in turn strengthen social capital. Furthermore, based on the survey results, respondents were least satisfied with the advertisement of the community center (the mean of satisfaction level is 2.85 based on the measurement scale 1–4). Therefore, promoting the benefits of the community center such as creating a flyer to post on the community board may be an effective approach.

Third, the association between chronic disease and social capital implies that those with chronic disease are more likely to connect with others in the community and enjoy a higher level of community belonging. Social capital is the resource that enhances residential interactions and networks. This social resource may play a key role in not only enhancing community belonging but also providing various health information among residents [44,45]. For example, social capital can contribute to the exchange of health information to enhance individual chronic disease self-management [46]. Hence, residents with chronic diseases are more likely to be associated with a higher level of social capital, thereby reducing their stress level. This result suggests that programs focusing on engaging residents with chronic illness should be given high priority in the community. The findings that high-income families and households with more children had a lower level of social capital suggest the importance of designing programs for high-income families and families with children which provide child activities or events in raising their social capital in the community. Moreover, respondents were not satisfied with the child-care programs provided by the community center, based on the survey results (the mean of satisfaction level is 2.96 based on the measurement scale 1–4). Consequently, obtaining input from families with children to design a satisfactory child-care program should be given special attention.

This study has significant results and implications, empirically identifying the nexus among multifaceted factors, social capital, and stress. Additionally, it is noteworthy that this study is the first empirical research based on the household drop-off survey for the REIP sites in Korea. However, there are some limitations that should be addressed. First, this study was based on a survey to measure the perceptions of physical environments, a methodology that may be subject to recall bias since responses are self-reported. Using an objective measure for physical environments might help address bias and provide an alternative approach to account for environmental features. Future studies should consider the influence of built environments using both subjective and objective measures and compare the potential differential associations with social capital and perceived stress. Second, this study included a limited number of physical environment variables such as walkability, transit accessibility, neighborhood amenities, crime safety, and cleanliness. Other dimensions such as perceived traffic safety in the community, particularly in terms of the provision of traffic calming devices and the presence of crosswalks, should be considered in a future study. Third, this survey was conducted during the COVID-19 pandemic in Korea. This unprecedented event may potentially result in recall biases for survey responses prior to the COVID-19 pandemic.

## 6. Conclusions

Although previous studies have explored the association between social capital and perceived stress, very few studies have focused on the mediating role of social capital on perceived stress. This study addressed this research gap by using the survey data from two neighborhoods that implemented the REIP in Seoul, Korea, and applied the SEM model to investigate this relationship. We found that multifaceted factors are related with perceived stress level through the mediator of social capital. Particularly, the role of social capital is substantial to reduce individual stress level, given the largest standardized coefficient of social capital in the empirical model. 

Our findings based on the structural framework have important policy implications for future study and practice. It is noteworthy that the frequency of community center use had both direct and indirect impacts on individual perceived stress through social capital. Beyond the financial support for building community centers, the public sector should make every effort to enhance community center uses in the REIP sites. The public sector should provide a tailored guideline for residents and the community committee on what programs and contents can be applied to enhance social capital and community health. For example, the provision of targeted programs for child-care and activities may increase the use of community centers, thereby strengthening social capital and reducing perceived stress. Furthermore, the collaboration between the public sectors, community committees, and public health practitioners needs to provide the opportunities and programs for exchanging various health care information for chronic disease self-management. Additionally, specific guidelines for enhancing community-based programs, including neighborhood cleanliness and maintenance, need to be developed to increase social capital and reduce individual stress. This approach may help residents by improving communities’ physical quality as well as strengthening social capital and mental health.

## Figures and Tables

**Figure 1 ijerph-19-05571-f001:**
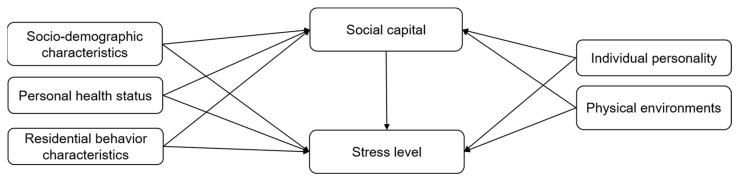
Proposed conceptual framework.

**Figure 2 ijerph-19-05571-f002:**
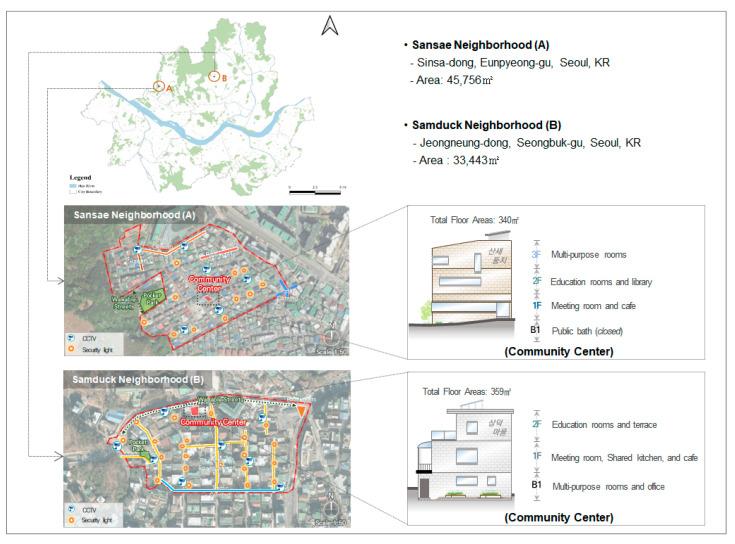
The location of community centers in Sansae and Samduck neighborhoods, Seoul.

**Figure 3 ijerph-19-05571-f003:**
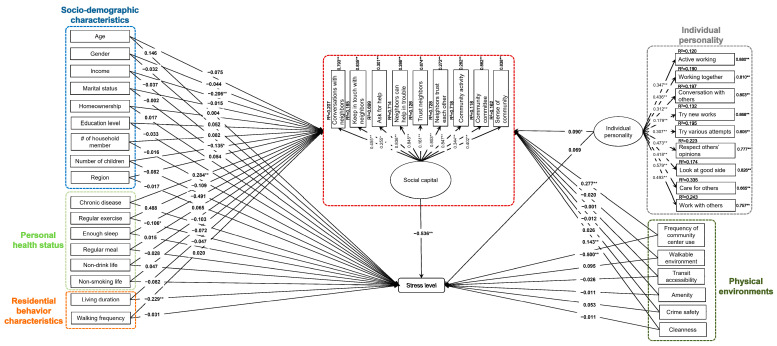
Standardized results of the structural equation model for this study. (**: *p* < 0.01; * *p* < 0.05).

**Table 1 ijerph-19-05571-t001:** Domains, variables, measurements, and descriptive statistics.

Domain	Variable	Measurement	Mean (SD ^a^) or % of “1” for Binary Variables
	Stress level	How much stress do you usually experience in your daily life? (1: no stress–4: very stressful)	2.68 (0.64)
**Social capital**	Latent factor: social capital	Network: Do you have many conversations with your neighbors? (1: never–4: always)	2.76 (0.58)
		Network: Do you often keep in touch with your neighbors? (1: never–4: always)	2.82 (0.65)
		Reciprocity: Do you have any neighbors you can ask for advice? (1: never–4: always)	2.95 (0.70)
		Reciprocity: Do you have any neighbors who can help you when you are in trouble? (1: never–4: always)	2.96 (0.71)
		Trust: Do you trust your neighbors? (1: never–4: always)	2.95 (0.62)
		Trust: Do you think neighbors trust each other in your neighborhood? (1: never–4: always)	2.89 (0.58)
		Participation: Do you participate in community activities? (1: never–4: always)	2.70 (0.67)
		Participation: Do you participate in community volunteer services? (1: never–4: always)	2.63 (0.73)
		Sense of belonging: Do you have a sense of community in the neighborhood? (1: never–4: always)	2.85 (0.53)
**Socio-demographic characteristics**	Age	Respondent’s age (years)	54.99 (10.59)
	Gender	Gender (0: female; 1: male)	26.76%
	Income	Household monthly income (1: below KRW 500,000 ^**b**^; 2: KRW 500,000–999,999; 3: KRW 1,000,000–1,499,999; 4: KRW 1,500,000–1,999,999; 5: KRW 2,000,000–2,499,999; 6: KRW 2,500,000–2,999,999; 7: KRW 3,000,000–3,499,999; 8: KRW 3,500,000–3,999,999; 9: KRW 4,000,000–4,499,999; 10: >= KRW 4,500,000)	7.56 (2.40)
	Marital status	Marital status (0: other status; 1: single)	1.54%
	Homeownership	Homeownership (0: no; 1: yes)	53.86%
	Education level	Education level (1: elementary; 2: middle; 3: high; 4: university; 5: graduate)	2.89 (0.76)
	Number of household members	Number of household members (continuous)	3.17 (0.91)
	Number of children	Number of children (continuous)	1.78 (0.58)
	Region	0: Samduck community; 1: Sansae community	71.53%
**Personal health status**	Chronic disease	Based on doctor’s diagnosis, do you have any chronic diseases? (0: no; 1: yes)	19.04%
	Regular exercise	Do you have regular physical exercises? (1: not often–4: very often)	3.29 (0.70)
	Enough sleep	Do you get enough rest and sleep? (1: not often–4: very often)	3.66 (0.66)
	Regular meal	Do you have a regular meal? (1: not often–4: very often)	3.77 (0.67)
	Non-drinking life	Do you have a non-drinking life pattern? (1: not often–4: very often)	3.68 (0.92)
	Non-smoking life	Do you have a non-smoking life pattern? (1: not often–4: very often)	3.87 (0.86)
**Residential behavior characteristics**	Living duration	Total living duration in the neighborhood (months)	165.46 (88.04)
	Walking frequency	Walk around neighborhoods over 10 minutes a week (1: none; 2: 1 day; 3: 2 days; 4: 3 days; 5: 4 days; 6: 5 days; 7: 6 days; 8: every day)	5.45 (1.67)
	Use frequency of community center	How often do you use the community center? (1: not often–5: very often)	1.67 (1.12)
**Individual personality**	Latent factor: individual personality	Extraversion: I like active working rather than passive working (1: strongly disagree–5: strongly agree)	3.57 (0.71)
		Extraversion: I like working together (1: strongly disagree–5: strongly agree)	3.77 (0.64)
		Extraversion: I like having a conversation with others in a group (1: strongly disagree–5: strongly agree)	3.74 (0.61)
		Openness: I like trying new works (1: strongly disagree–5: strongly agree)	3.69 (0.73)
		Openness: I tend to try various attempts to solve the issue (1: strongly disagree–5: strongly agree)	3.60 (0.65)
		Openness: I try to respect other people’s different opinions (1: strongly disagree–5: strongly agree)	3.69 (0.61)
		Agreeableness: I try to look at the good side of others (1: strongly disagree–5: strongly agree)	3.72 (0.66)
		Agreeableness: I try to care for others (1: strongly disagree–5: strongly agree)	3.80 (0.59)
		Agreeableness: I like the collaboration with others rather than the competition with others (1: strongly disagree–5: strongly agree)	3.74 (0.59)
**Physical environments**	Walkable neighborhood	How satisfied are you with walkable environments in your neighborhoods? (1: not satisfied–4: very satisfied)	2.98 (0.60)
	Transit accessibility	How satisfied are you with transit accessibility in your neighborhoods? (1: not satisfied–4: very satisfied)	2.85 (0.66)
	Amenity	How satisfied are you with the amenity in your neighborhoods? (1: not satisfied–4: very satisfied)	2.97 (0.68)
	Crime safety	How satisfied are you with the crime safety in your neighborhoods? (1: not satisfied–4: very satisfied)	2.77 (0.61)
	Cleanliness	How satisfied are you with the cleanliness in your neighborhoods? (1: not satisfied–4: very satisfied)	2.90 (0.39)

^**a**^ SD: standard deviation; ^**b**^ KRW 1000 = about USD 0.9 as of 18 May 2021.

**Table 2 ijerph-19-05571-t002:** The relationships between socio-demographic characteristics, social capital, and stress level.

	Social Capital		Stress Level	
	Standardized Coefficient	*p*-Value	Standardized Coefficient	*p*-Value
Sociodemographic Attributes				
Age	−0.075	0.231	0.146	0.318
Gender	−0.044	0.112	−0.032	0.293
Income	−0.206 **	0.002	−0.037	0.221
Marital status	−0.015	0.548	−0.002	0.843
Homeownership	0.004	0.085	0.017	0.624
Education level	0.052	0.372	−0.033	0.549
Number of household members	0.082	0.424	−0.016	0.387
Number of children	−0.135 *	0.015	−0.082	0.392
Region	0.054	0.322	−0.017	0.228

**: *p* < 0.01; * *p* < 0.05.

**Table 3 ijerph-19-05571-t003:** The direct and indirect relationships between personal health status, residential behavior characteristics, social capital, and stress level.

	Direct Impact				Indirect Impact
	Social Capital		Stress Level		Stress Level
	Standardized Coefficient	*p*-Value	Standardized Coefficient	*p*-Value	
Personal health status					
Chronic disease	0.284 **	0.003	0.488	0.432	−0.152 **
Regular exercise	−0.109	0.176	−0.106 *	0.031	-
Enough sleep	−0.491	0.144	0.015	0.434	-
Regular meal	0.065	0.284	−0.028	0.842	-
Non-drinking life	−0.103	0.743	0.047	0.648	-
Non-smoking life	−0.072	0.695	−0.082	0.553	-
Residential behavior characteristics					
Living duration	−0.047	0.386	−0.229 **	0.002	-
Walking frequency	0.020	0.245	−0.031	0.248	-
Use frequency of community center	0.277 **	0.003	−0.500 **	0.004	−0.148 **

**: *p* < 0.01; * *p* < 0.05.

**Table 4 ijerph-19-05571-t004:** The direct and indirect relationships between individual personality, physical environments, social capital, and stress level.

	Direct Impact				Indirect Impact
	Social Capital		Stress Level		Stress Level
	Standardized Coefficient	*p*-Value	Standardized Coefficient	*p*-Value	
Individual personality	0.090 **	0.003	0.069	0.072	−0.048 *
Physical environments					
Walkable environment	−0.020	0.080	0.095	0.194	-
Transit accessibility	−0.001	0.132	−0.026	0.145	-
Amenity	−0.012	0.096	−0.011	0.133	-
Crime safety	0.026	0.144	0.053	0.241	-
Cleanness	0.143 **	0.005	−0.011	0.295	−0.077 **

**: *p* < 0.01; * *p* < 0.05

## Data Availability

The data presented in this study are available in the paper.

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
