# Peer review of "Effects of Multifaceted Determinants on Individual Stress: The Mediating Role of Social Capital"

_ijerph, 2022, doi:10.3390/ijerph19095571_

Round 1
Reviewer 1 Report
Beneficial text, supported by grants. Ethical principles adhered to. Literary review documented. Operationalization of terms described. Methodologically well anchored research. Significant application in practice.
However, it is absolutely necessary
- document some definitions and differences between them.
- restructure the text as a whole (information is included all - sometimes even redundantly)
- 1: explain terms: social control; networks; sociability; but especially the concept of social environmental and the difference between this concept and the concept of social capital (these are explained in the text)
Keywords behind the abstract to be more precise: the method of data processing is not one of the key terms
ad 2: In the Introduction it is necessary to state the problem to be solved, the definition and the main goal (research question, hypothesis). Not the literature review, which is in the next chapter. Everything overlaps. Maybe in the Introduction the project could be described and also the geographics part of Seoul - simply a problem to be solved. In the literature review at least indicate the search strategies.
The paragraph Discussion and Conclusion overlaps: interesting results need to be included in the discourse, a more concise discussion needs to be made and comparisons with similar surveys need to be made here. In Conclusion just put the application into practice and other possible ways of such research. In principle, it is clear - no results (and no text) to repeat.
Beneficial text, supported by grants. Ethical principles adhered to. Literary review documented. Operationalization of terms described. Methodologically well anchored research. Significant application in practice.
However, it is absolutely necessary
- document some definitions and differences between them.
- restructure the text as a whole (information is included all - sometimes even redundantly)
- 1: explain terms: social control; networks; sociability; but especially the concept of social environmental and the difference between this concept and the concept of social capital (these are explained in the text)
Keywords behind the abstract to be more precise: the method of data processing is not one of the key terms
ad 2: In the Introduction it is necessary to state the problem to be solved, the definition and the main goal (research question, hypothesis). Not the literature review, which is in the next chapter. Everything overlaps. Maybe in the Introduction the project could be described and also the geographics part of Seoul - simply a problem to be solved. In the literature review at least indicate the search strategies.
The paragraph Discussion and Conclusion overlaps: interesting results need to be included in the discourse, a more concise discussion needs to be made and comparisons with similar surveys need to be made here. In Conclusion just put the application into practice and other possible ways of such research. In principle, it is clear - no results (and no text) to repeat.
Reviewer 2 Report
Dear authors, thank you for giving me the opportunity to review your manuscript: “Effects of Multifaceted Determinants on Individual Stress: The Mediating Role of Social Capital”. I think that this manuscript can meaningfully contribute to the literature.
Introduction
The introduction presents a clear relationship between the variables and is congruent with the study's objectives.
I suggest the following adjustments:
Materials and Methods
- All ethical considerations should be explained.
- The inclusion criteria should be addressed.
- Has the research been submitted to and approved by the Ethics Committee? Which one?
Discussion
- These results should be discussed further in the discussion:
- “For personal health status, those who had chronic disease had a higher level of social capital.”,
- “Those who had chronic disease were associated with a lower stress level through social capital”
